# Rapamycin suppresses inflammation and increases the interaction between p65 and IκBα in rapamycin-induced fatty livers

Chenliang Ge[1], Changguo Ma[1], Jiesheng Cui[2], Xingbo Dong[2], Luyang Sun[3], Yanjiao Li[1], An Yu[1]*

1 Yunnan Key Laboratory for Basic Research on Bone and Joint Diseases & Yunnan Stem Cell Translational Research Center, Kunming University, Kunming, Yunnan, China, 2 Fabulous Artificial Intelligence Co., Ltd., Shenzhen, P.R. China, 3 Huffington Center on Aging, Baylor College of Medicine, Houston, TX, United States of America

* anyu@kmu.edu.cn

**Data Availability Statement:** All relevant data are within the paper and its Supporting Information files.

## Abstract

Rapamycin treatment significantly increases lifespan and ameliorates several aging-related diseases in mice, making it a potential anti-aging drug. However, there are several obvious side effects of rapamycin, which may limit the broad applications of this drug. Lipid metabolism disorders such as fatty liver and hyperlipidemia are some of those unwanted side effects. Fatty liver is characterized as ectopic lipid accumulation in livers, which is usually accompanied by increased inflammation levels. Rapamycin is also a well-known anti-inflammation chemical. How rapamycin affects the inflammation level in rapamycin-induced fatty liver remains poorly understood. Here, we show that eight-day rapamycin treatment induced fatty liver and increased liver free fatty acid levels in mice, while the expression levels of inflammatory markers are even lower than those in the control mice. Mechanistically, the upstream of the pro-inflammatory pathway was activated in rapamycin-induced fatty livers, however, there is no increased NFκB nuclear translocation probably because the interaction between p65 and IκBα was enhanced by rapamycin treatment. The lipolysis pathway in the liver is also suppressed by rapamycin. Liver cirrhosis is an adverse consequence of fatty liver, while prolonged rapamycin treatment did not increase liver cirrhosis markers. Our results indicate that although fatty livers are induced by rapamycin, the fatty livers are not accompanied by increased inflammation levels, implying that rapamycin-induced fatty livers might not be as harmful as other types of fatty livers, such as high-fat diet and alcohol-induced fatty livers.

## Introduction

Rapamycin has been reported to delay aging and extend mouse lifespan [1, 2]. It has been suggested that rapamycin should be used as an anti-aging drug to combat aging-related diseases [3, 4]. However, rapamycin has many unwanted side effects [5], such as glucose intolerance, hyperlipidemia [6], and fatty liver [7]. These unwanted side effects might limit the applications

**Funding:** This work was supported by the funding of Yunnan Key Laboratory for Basic Research on Bone and Joint Diseases (BRBJD-2021-1) and the funding of Kunming University (XJ20220008) awarded to An Yu. Yunnan Key Laboratory for Basic Research on Bone and Joint Diseases is a lab of Kunming University. The URL of Kunming University is https://www.kmu.edu.cn/ No, the sponsors did not play any role in the study design

**Competing interests:** The authors have declared that no competing interests exist.

of rapamycin as an anti-aging drug in humans. Fatty liver can be induced by several factors, including a high-fat diet, alcohol, and some drugs; in mouse models, fatty liver can also be induced by genetic manipulation such as hepatocyte-specific SIRT1 deletion [8] and hormone treatment such as resistin injection [9]. Common features of these different types of fatty livers are significantly increased lipid accumulation and increased inflammation levels in the livers. As a potential anti-aging drug, on one hand, rapamycin induces fatty liver [7]; on the other hand, rapamycin is a potent anti-inflammation drug [10–12]. How rapamycin affects the inflammation level in the fatty liver induced by itself remains unknown.

Nuclear factor kappa B (NFκB) is a master regulator of pro-inflammatory response and it is a dimeric transcription factor comprised of five family members RelA (p65), RelB, c-Rel, p50, and p52 [13]. The major form of NFκB is the p65-p50 dimer, and the activation of canonical NFκB pathway involves phosphorylation and activation of inhibitor of NF-κB (IκBα) kinase (IKK), then IκBα was phosphorylated by IKK and subsequently degraded by ubiquitin-proteasome pathway resulting in NFκB nuclear translocation [14]. Since IκBα retains NFκB in the cytoplasm, IκBα prevents NFκB from being a functional transcriptional factor. Therefore, phosphorylation and degradation of IκBα are thought to be key steps to activate the NFκB pathway. In fatty livers, increased free fatty acid levels have been considered a major adverse factor causing hepatocyte dysfunction [15]. Mechanistically, high levels of free fatty acids can activate the NFκB pathway through Toll-like receptor (TLR) 2 and 4 [16]. It has been reported that, in rapamycin-induced fatty liver, the levels of free fatty acids were significantly increased [7]. Because activation of the NFκB pathway plays an important role in the development of liver fibrosis from the fatty liver [17], determining how rapamycin affects the NFκB pathway in rapamycin-induced fatty liver is necessary to assess the undesired side effects of rapamycin as an anti-aging drug. It was observed that free fatty acid levels were increased in the rapamycin-induced fatty liver [7] and saturated free fatty acids are pro-inflammatory factors, which are usually released from lipolysis, therefore, how rapamycin affects the lipolysis pathways in the liver is also an interesting question to be investigated.

In the present study, we found that eight-day rapamycin treatment induced fatty livers and significantly increased liver free fatty acid levels in mice; however, the expression levels of inflammation markers are lower in rapamycin-induced fatty livers than those in the control group. The interaction between p65 and IκBα was enhanced by rapamycin treatment resulting in unchanged p65 nuclear translocation despite the factor that the phosphorylation levels of IKK were increased and IκBα levels were decreased in rapamycin-induced fatty livers. While in high fat diet-induced fatty livers, there is a significantly increased p65 nuclear translocation. In addition, even though the free fatty acid levels were increased in rapamycin-induced fatty livers, G0/G1 switch gene 2 (G0S2), which is the inhibitor of adipose triglyceride lipase (ATGL), was significantly upregulated in rapamycin-induced fatty livers, suggesting that rapamycin may restrict lipolysis in the liver. Although fatty liver has been considered a risk factor for liver fibrosis/cirrhosis development, prolonged rapamycin treatment did not increase liver fibrosis markers. Our results imply that rapamycin-induced fatty livers might not be as harmful as other types of fatty livers, such as high fat diet-induced fatty livers.

## Materials and methods

### Materials

DMEM/F12 media was purchased from Hyclone and fetal bovine serum (FBS) was purchased from Gibco (Beijing, China). TNF-α, COX-2, and p-IKK antibodies were purchased from Cell Signaling Technology (Beverly, MA). β-actin, p65, IκBα, IL-1β, and Caspase-1 antibodies were purchased from Abcam (HKSP, N.T. Hong Kong). CHOP antibody was purchased from Santa

Cruz Biotechnology, Inc. (Santa Cruz, CA). Rapamycin, DMSO, and all other chemicals were purchased from Sigma (Shanghai, China).

## Animals and isolation of mouse primary hepatocytes

12 weeks old male C57BL/6 mice were housed in a temperature-controlled environment and maintained on a 12 h light/dark cycle. Rapamycin stock solution: dissolve 10 mg of rapamycin in 187.5 μL DMSO, then the concentration is 0.05 mg/μL. Take 48 μL rapamycin stock solution and add it to the mixture of DMSO and 0.9% normal saline (50% DMSO and normal saline) to dilute rapamycin, making it to rapamycin working solution. Eight mice were treated with rapamycin (80 μg/mouse/day) and administered daily intraperitoneal injection for eight days. The prolonged treatment is a daily intraperitoneal injection for eighteen weeks. Seven mice of the same age were injected with PBS/DMSO as the control group. Mice fasted overnight before sacrifice and collection of liver tissue. We used Carbon Dioxide (CO2) to euthanize both adult and newborn mice. CO2-mediated euthanasia provides a rapid, painless, stress-free death because CO2 overdose causes rapid unconsciousness followed by death. Mice that need to be sacrificed were put into the euthanasia chamber and the $CO_2$ flow rate displaces 30% of the chamber volume per minute. Gas flow was maintained for at least 1 minute after apparent clinical death. Because we did not carry out surgeries on mice, no anesthesia or analgesia was needed. The effect of rapamycin on mTORC1 suppression was indicated by the phosphorylation level of S6 protein (S1A Fig). Mouse primary hepatocytes were isolated as previously indicated [18]. Briefly, livers from newborn C57BL/6 mice were sliced and then digested with a mixture of 0.125% trypsin and 0.1% collagenase II for 30 min at 37˚C. After digestion, trypsin and collagenase were removed by centrifugation, and pellets were fully re-suspended in 10 mL fetal bovine serum. Hepatocytes were purified by maintaining the serum re-suspended pellets at 4˚C for 40 min then spinning briefly. After spinning, the serum was removed, and hepatocytes were re-suspended in 10% FBS DMEM/F12 medium. Mouse high fat diet (HFD) was purchased from Xietong Pharmaceutical Bio-engineering Co., Ltd. (Jiangsu, China). Mice were fed with HFD for two months to induce fatty livers. The laboratory procedures conformed to the Kunming University guidelines for the Care and Use of Laboratory Animals (permit numbers KMUSM20210903).

## Quantitative real-time PCR (qPCR) and western blotting

2 μg total RNA of each sample was used to perform reverse transcription. qPCR was carried out in a total volume of 20 μL with SYBR-Green mix (Takara, Dalian, China) on an IQ5 thermal cycler (Bio-Rad, Hercules, CA). Mouse 36B4 was used as the internal reference and amplified in parallel. The PCR conditions were 95˚C for 3 min, followed by 40 cycles of 95˚C for 15 s, 60˚C for 15 s, and 72˚C for 15 s. Cycle threshold values were normalized to that of the internal references, and the relative gene expression levels were calculated by the $2^{-\Delta\Delta Ct}$ method. The sequences of all qPCR primers are shown in the S1 Table.

Proteins were extracted by RIPA buffer. Aliquots containing 60 μg of protein from each sample were separated by 10% SDS–PAGE then all proteins were transferred onto polyvinylidene difluoride membranes. The membranes were blocked with TBST buffer (20 mM Tris–HCl, 137 mM NaCl, and 0.05% Tween-20) containing nonfat dried milk at room temperature for 60 min then incubated overnight with a 1:1000 dilution of appropriate primary antibodies. After extensive washing, the membranes were incubated with horseradish peroxidase-conjugated secondary antibodies (Santa Cruz; 1:4000) for 1 h at room temperature and visualized using an ECL Western blotting detection system (Tiangen, Beijing, China).

### Gene microarray, immunostaining, and free fatty acids test

200 μg RNA of each sample was provided to WUHAN EASYDIAGNOSIS BIOMEMDICINE Co., Ltd. (Wuhan, China) to perform gene microarray analysis. GO term and KEGG analysis is performed by using R with the DAVID package. GO-terms and KEGG pathways with fold enrichment FDR p-value < 0.05 were considered significantly enriched. The Venn diagram was generated by https://bioinfogp.cnb.csic.es/tools/venny/index.html. Oil red O staining and p65 immunostainings were achieved by Proteintech Group, Inc. (Wuhan, China). Liver free fatty acids were analyzed by a free fatty acid kit (Abcam).

### Statistical analysis

All experiments were repeated at least three times with similar results. Data are presented as means ± SD. Student's t-test was used for statistical comparison. P < 0.05 was considered statistically significant.

## Results

### Rapamycin induced fatty livers in mice

To confirm previous results that fatty liver can be induced by rapamycin [7], 12-week-old male mice were treated with rapamycin (80 μg/mouse/day) for 8 days. Lipid accumulation can be directly reflected by the color of mouse livers upon rapamycin treatment (Fig 1A). Oil red O staining in mouse liver frozen sections indicates that 8-day rapamycin treatment induced more lipid accumulation than the lipid accumulation caused by two months high fat diet (HFD)in mouse livers (Figs 1B and S2F). Although it was thought that rapamycin treatment may mimic nutrient deprivation, microarray analysis revealed that the mRNA expression patterns in the livers of rapamycin treatment mice and 24 h fasting mice are quite different (Fig 1C). Both fasting and rapamycin treatment are able to inhibit mTOR activity [19], therefore rapamycin does mimic some aspects of fasting; however, in our current study, we found that rapamycin treatment and 24 h fasting-induced very different gene expression patterns (Fig 1C). The reasons behind this phenomenon could be that rapamycin mainly inhibits mTOR signaling whereas fasting affects many pathways including Sirts (Sirt1-Sirt7), AMPK, and mTOR signaling. The GO function and KEGG pathway analysis of microarray also revealed that lipid metabolism genes in livers are significantly altered by rapamycin treatment (Figs 1D and 1E and 2A and 2B). The numbers of differentially expressed genes in the livers of rapamycin-treated mice and fasting mice were shown in the Venn diagram (Fig 1F). To verify the RNA microarray results, quantitative PCR (qPCR) was performed to determine the mRNA expression levels of lipogenesis genes and lipid catabolism genes in the livers of rapamycin-treated mice. Indeed, key lipogenesis genes (ACC1: acetyl-coenzyme A carboxylase α; SREBP-1c: sterol regulatory element binding transcription factor 1 c; FASN: fatty acid synthase) were significantly upregulated by rapamycin (Fig 2C); whereas, fatty acid β-oxidation enzymes (ACOX-1: acyl-coenzyme A oxidase 1; CPT-1α: carnitine palmitoyltransferase 1α) were significantly downregulated (Fig 2D). The free fatty acids importer CD36 and lipoprotein transporter microsomal triglyceride transfer protein (Mttp) expression levels were not significantly affected by rapamycin (Fig 2E). And the mRNA expression levels of other key proteins that may affect lipid transport in livers were also determined by Q-PCR, the results indicated that there are no dramatic expression changes for these genes upon rapamycin treatment (S2D Fig). Consistent with previous results [7], free fatty acids level was significantly increased in rapamycin induced fatty livers (Fig 2F). These results suggest that rapamycin increased lipid

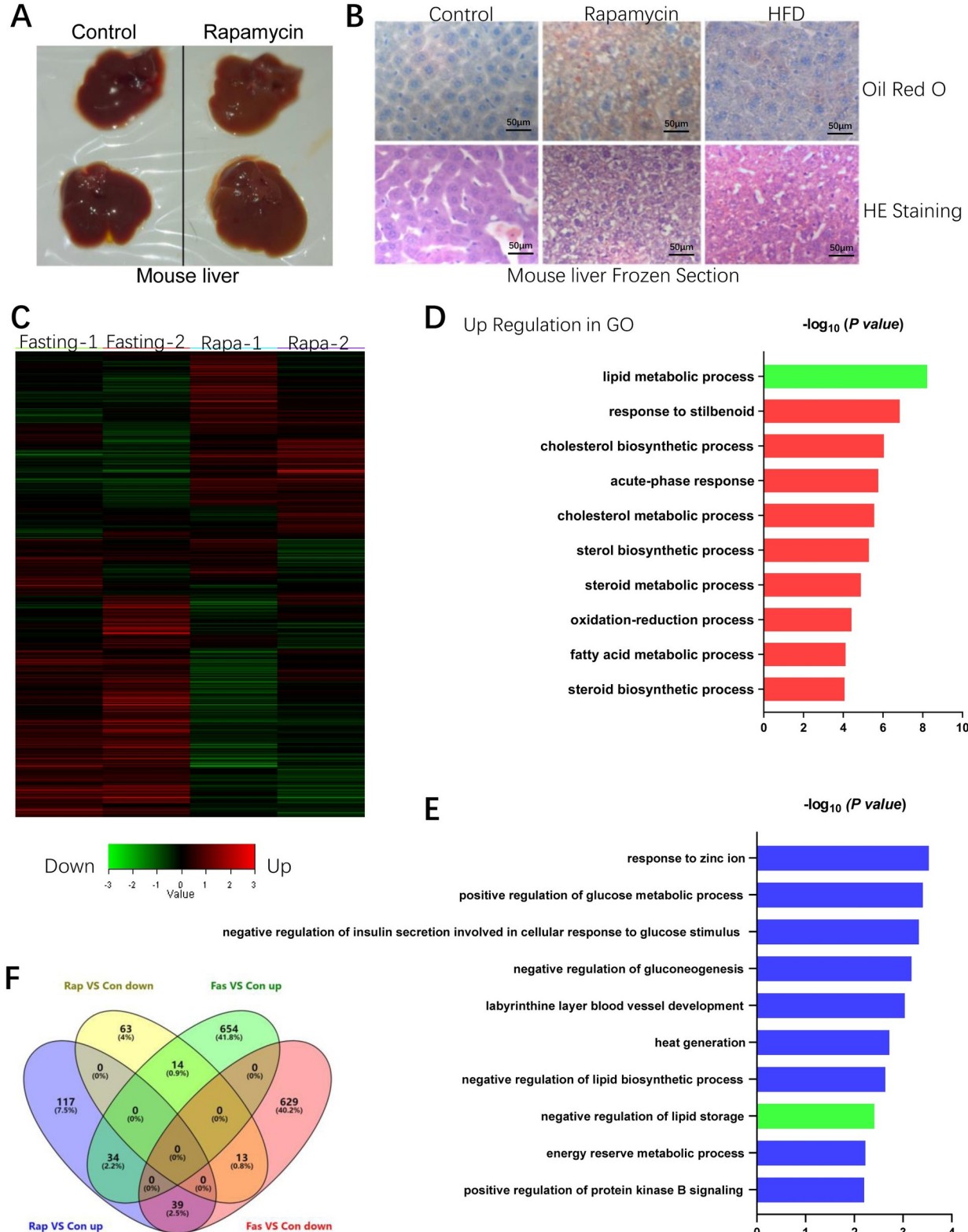

**Fig 1. Rapamycin-induced lipid accumulation in mouse liver.** (Control group n = 7, rapamycin treated group n = 8; Data represent mean ± SD, **p < 0.01, *p<0.05). A. Mouse livers of rapamycin-treated group and control group. B. Oil red O staining indicates remarkable lipid accumulation in rapamycin-induced fatty livers. HFD group: mice were fed with the high fat diet for two months. C. Microarray analysis revealed

the liver gene expression patterns are quite different between rapamycin-treated mice and 24 h fasting mice. D & E. GO term enrichment analysis reveals that rapamycin in the mouse liver significantly affects lipid metabolism (highlighted in green). F. Differentially expressed genes in rapamycin-treated mice and fasting mice were indicated by the Venn diagram.

accumulation in livers via upregulated lipogenesis pathway but not the increased lipid absorption nor the decreased lipoprotein export.

## Rapamycin suppresses pro-inflammatory NFκB signaling by increasing p65 and IκBα interaction

Since increased free fatty acid levels were observed in rapamycin-induced fatty livers (Fig 2F) and free fatty acids are potent pro-inflammatory factors [20], whether increased inflammation levels can be caused by increased free fatty acids becomes a question. Western blotting of inflammation markers indicated that there is an obvious downregulation of p65 and tumor necrosis factor α (TNF-α) in rapamycin-treated livers (Figs 3A and S1B), suggesting that the inflammation levels of rapamycin-induced fatty livers are even lower than that in the control livers. In mouse primary hepatocytes, rapamycin treatment did not decrease the p65 protein levels (Figs 3B and S1C) suggesting that the way in which rapamycin affects liver tissue is not identical to the way in hepatocytes. In high fat diet (HFD)-induced fatty livers, the phosphorylation levels of IKK, which indicates the activation of IKK, were dramatically increased; and the protein levels of IκBα were decreased (Figs 3C and S1D). These results suggest the activation of NFκB pathway. Surprisingly, in rapamycin-induced fatty livers, the increased phosphorylation levels of IKK and decreased protein levels of IκBα were also observed (Figs 2D and S1E), indicating that there are steps of activation of the NFκB pathway in rapamycin-induced fatty livers. Since IκBα functions as a key inhibitor of NFκB by retaining p65-p50 dimer in the cytoplasm, decreased IκBa protein levels could participate in facilitating NFκB nuclear translocation that can be visualized by p65 immunostaining; to assess the degree of NFκB nuclear translocation, p65 immunostaining was performed in the frozen sections of control mouse livers, HFD-induced fatty livers, and rapamycin-induced fatty livers. The results showed that p65 nuclear translocation was significantly increased in HFD induced fatty liver whereas the nuclear translocation of p65 in the rapamycin-induced fatty liver was not changed compared to the control mouse livers (Fig 3E). These results indicated that although IκBα protein levels were decreased in both HFD-induced fatty livers and rapamycin-induced fatty livers, NFκB nuclear translocation was increased only in HFD-induced fatty livers. This result could be understood as there might be underlying mechanisms inhibiting NFκB nuclear translocation despite the lower IκBα protein levels in rapamycin-induced fatty livers. Indeed, co-immunoprecipitation experiments showed that the interaction between p65 and IκBα was enhanced in rapamycin-induced fatty livers (Figs 3F and S1F), therefore NFκB was retained in the cytoplasm (Fig 3E), even the upstream of NFκB pathway was activated (Fig 3D). In addition to NFκB pathway activation by extracellular stimulation, endogenous NOD-like receptor (NLR) family, pyrin domain-containing protein 3 (NLRP3) inflammasome also largely contributes to inflammation process. The NLRP3 inflammasome is composed of NLRP3, Caspase-1, and the adaptor protein ASC, and blocking voltage-dependent anion channels (VDAC1 and 2) is able to suppress the activation of NLRP3 inflammasome [21]. In rapamycin-induced fatty livers, the mRNA levels of NLRP3, Caspase-1, and VDAC2 were significantly downregulated (Fig 3G). As has been shown in S3 Fig, rapamycin treatment significantly decreased IL-1β protein levels in mouse livers (S3A and S3B Fig), and notably, the cleaved IL-1β for secretion was hardly detectable in rapamycin-induced fatty livers, though another cleaved IL-1β protein levels were not affected by rapamycin (S3C Fig). Also, rapamycin treatment significantly decreased Caspase-1 protein levels (S3D–

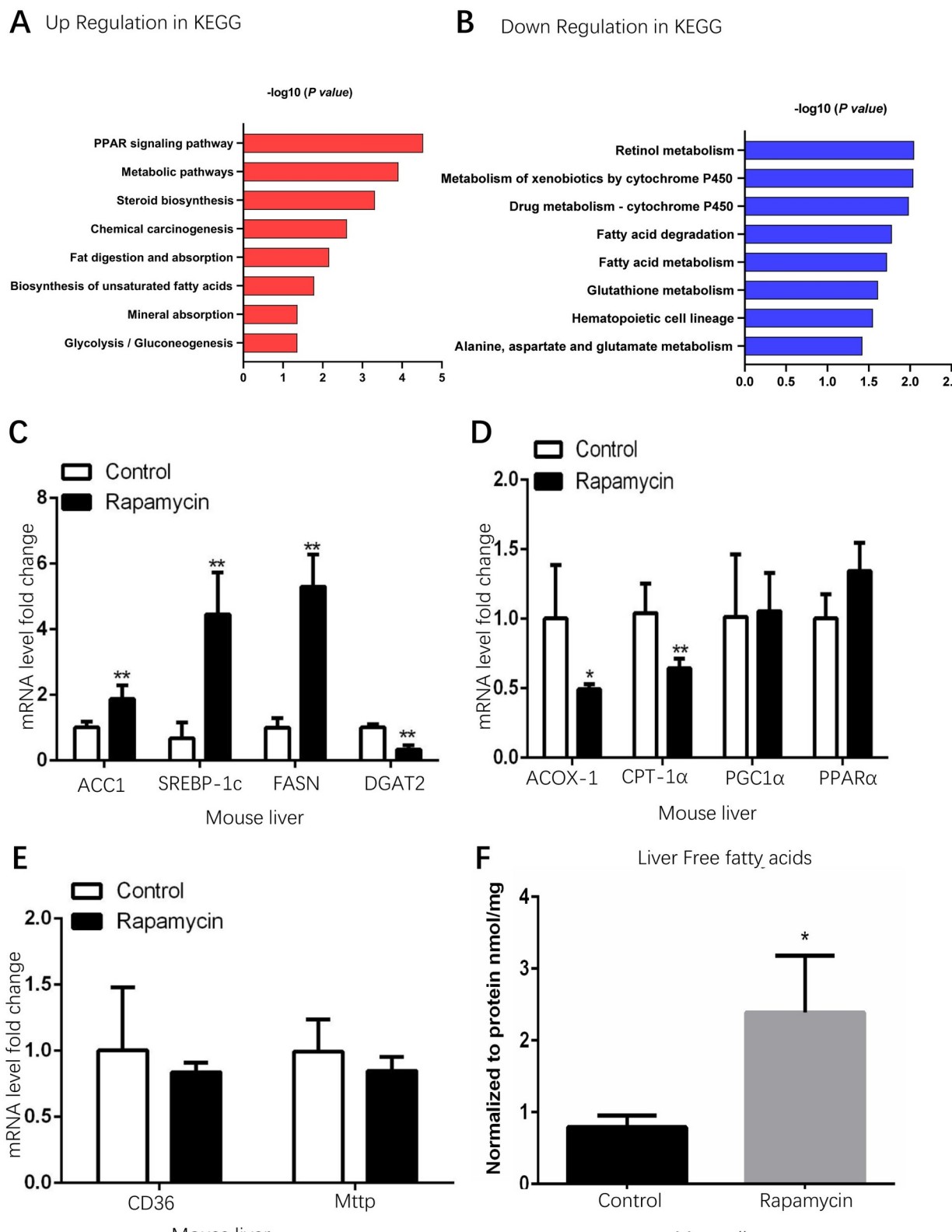

**Fig 2. Rapamycin increased lipogenesis gene expressions in mouse liver.** (Control group n = 7, rapamycin treated group n = 8; Data represent mean ± SD, $^{**}$p < 0.01, $^{*}$p<0.05). A & B. KEGG pathway analysis reveals that fatty acid metabolism is significantly affected by rapamycin in mouse liver. C. lipogenesis genes were significantly upregulated in the livers of rapamycin-treated mice. D. Moderate downregulation of fatty

acids oxidation genes by rapamycin in mouse livers. E. Fatty acids importer CD36 and lipoprotein exporter Mttp were not affected by rapamycin. F. Free fatty acids level was significantly increased in rapamycin-induced fatty liver.

S3F Fig), though it is hard to determine whether cleaved Caspase-1 levels were affected by rapamycin due to the very low signaling (S3F Fig). These results indicate that NLRP3 inflammasome activation may be suppressed by rapamycin at least via decreasing Caspase-1 and IL-1β expression levels and reducing cleaved IL-1β form for secretion. Taken together, these results indicate that although the pro-inflammatory free fatty acids were increased in rapamycin-induced fatty livers (Fig 2F), simultaneously, rapamycin is able to suppress the inflammation response through downregulating NLRP3 inflammasome expression and enhancing the interaction between p65 and IκBα.

## Rapamycin upregulates lipolysis inhibitor G0S2 in livers

Adipose triglyceride lipase (ATGL) is the key enzyme for the release of fatty acids from triglycerides (TG), while G0/G1 switch gene 2 (G0S2) is its potent inhibitor and comparative gene identification-58 (CGI-58) is its activator [22]. Since increased free fatty acids were found in rapamycin-induced fatty livers (Fig 2F), the expressions of lipolysis-associated genes were determined in rapamycin-treated mice. The qPCR results showed that in the livers of rapamycin-treated mice, ATGL and CGI-58 were downregulated while G0S2 was very significantly upregulated (Fig 4A). The upregulation of G0S2 was not observed in the adipose tissues of rapamycin-treated mice (Fig 4B). The western blotting experiments confirmed that the upregulation of G0S2 by rapamycin can only be detected in livers (Figs 4C and S2A) but not in adipose tissues (Figs 4D and S2B). These results suggest that rapamycin has a liver-specific effect on G0S2 expression and the increased free fatty acid levels (Fig 2F) are not due to the activation of lipolysis pathway in rapamycin-induced fatty livers. Besides, the increased mRNA expression level of DGAT2 (diacylglycerol O-acyltransferase 2) was observed in the livers of rapamycin-treated mice (S2E Fig), implying that increased free fatty acid levels might be due to the increased lipogenesis (Fig 2C) instead of the decreased esterification reaction.

Endoplasmic reticulum (ER) stress plays an important role in the pathogenesis of fatty livers [23], the ER stress-associated genes were also tested in rapamycin-induced fatty livers. The results showed that there are no significant expression changes of ER stress genes (Fig 4E). However, the C/EBP homologous protein (CHOP), which plays an important role in ER stress-induced apoptosis [24], was increased in rapamycin-induced fatty livers (Figs 4F and S2C). This result suggests that even if there are no increased inflammation levels, other cellular stress might be induced by ectopic lipid accumulation in rapamycin-induced fatty livers. Fatty liver has been considered a risk factor for liver fibrosis/cirrhosis development [25], eighteen-week rapamycin treatment in mice did not increase the expression of liver fibrosis markers [26] (Adamts1 and TGF-β1) (Fig 4G), suggesting that rapamycin-induced fatty livers may not be a risk factor for the development of liver cirrhosis. Collectively, our results imply that rapamycin-induced fatty livers might not be as harmful as other types of fatty livers, such as high fat diet-induced fatty livers.

## Discussion

Saturated fatty acids in obesity can induce chronic inflammation and have harmful effects on multiple organs and systems [27]. In fatty livers, free fatty acids activate the NFκB pathway to increase inflammation levels [16]. Rapamycin has been suggested to be an anti-inflammation reagent during the last few years, for example, in chondrocytes, rapamycin inhibits NFκB

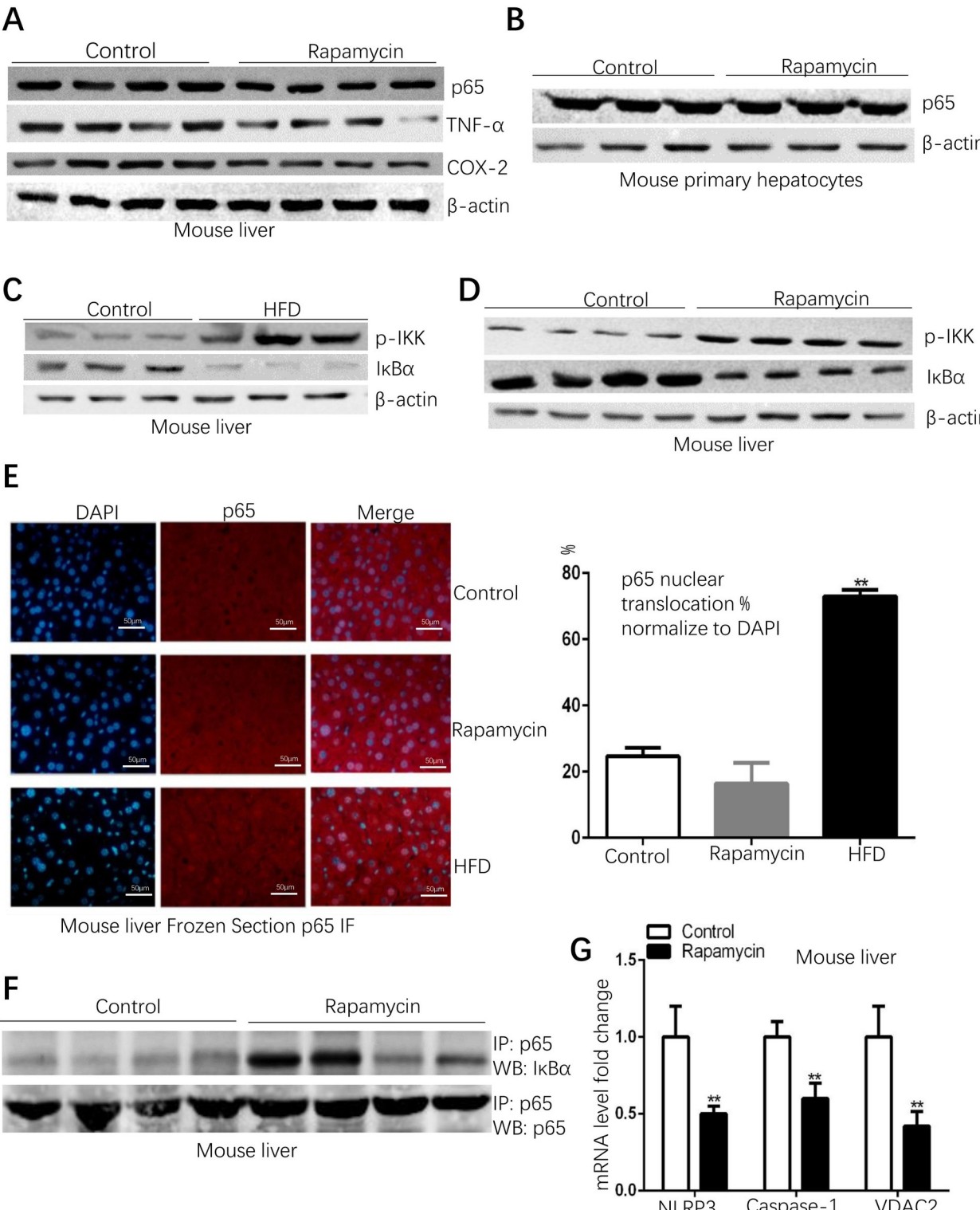

**Fig 3. Rapamycin enhances the interaction between p65 and IκBα to block the NFκB pathway.** (Data represent mean ± SD, $^{**}$p < 0.01, $^{*}$p<0.05). A. Lower expressions of pro-inflammatory genes in rapamycin-induced fatty liver. (Control group n = 4, rapamycin group n = 4). B. p65 protein level was not changed in mouse primary hepatocytes. C. Increased phosphorylation level of IKK and decreased IκBα level in high fat diet (HFD)-induced fatty livers (Control group n = 3, HFD group n = 3). D. Increased phosphorylation level of IKK and decreased IκBα level in rapamycin-induced fatty livers (Control group n = 4, rapamycin group n = 4). E. Increased p65 nuclear translocation in HFD-induced fatty livers; p65 nuclear translocation was not increased in rapamycin-induced fatty livers. F. Interaction between p65 and IκBα has enhanced in rapamycin-

induced fatty livers (Control group n = 4, rapamycin group n = 4). G. NLRP3 inflammasome-related genes were downregulated in rapamycin induce fatty livers (Control group n = 7, rapamycin-treated group n = 8).

activation by increasing autophagy [28]; in senescent cells, rapamycin inhibits NFκB activation by suppressing interleukin-1α translation [11]. Rapamycin-induced fatty livers provide a unique model to study how rapamycin affects the inflammation levels in fatty livers because the pro-inflammatory free fatty acids were increased by rapamycin itself (Fig 2F), which has been indicated by the previous study [7]. In the present study, despite the increased free fatty acid levels (Fig 2F), we found that the expression levels of pro-inflammatory markers are even lower in the rapamycin-induced fatty livers than those in control mouse livers (Figs 3A and S1B). This observation generated a paradox that requires an explanation of the reason why the increased fatty acid levels did not provoke the pro-inflammatory response in rapamycin-induced fatty livers. Indeed, the upper stream factors of the NFκB pathway, which is one of the major pathways regulating pro-inflammatory response, were activated in rapamycin-induced fatty livers (Fig 3D). However, the next step of NFκB activation, which is the increased p65 nuclear translocation, was not observed (Fig 3E). These results suggest that there might be underlying mechanisms suppressing the p65 nuclear translocation despite the decreased IκBα protein levels in rapamycin-induced fatty livers (Fig 3D). Notably, we found a previously unnoticed anti-inflammation mechanism of rapamycin (Fig 4H), which is to enhance the interaction between p65 and IκBα by rapamycin (Figs 3F and S1F). Considering that the enhanced interaction between p65 and IκBα may help IκBα to restrict the p65 nuclear translocation by retaining p65 in cytoplasm, this mechanism may help to explain the reason why although IKK was activated and IκBα protein levels were decreased (Figs 3D and S1E), the p65 nuclear translocation was not increased in rapamycin-induced fatty livers (Fig 3E). In addition, rapamycin seems to suppress NLRP3 inflammasome activation by down-regulating the expression levels of VDAC2, NLRP3 and Caspase-1 (Figs 3G and S3D–S3F). The suppressed NLRP3 inflammasome activation can be indicated by significantly decreased IL-1β protein levels (S3A and S3B Fig) and dramatically reduced cleaved IL-1β for secretion (S3C Fig). These results suggest that rapamycin functions as a potent anti-inflammatory drug by multiple mechanisms and is able to counteract the potential pro-inflammatory effects caused by itself, such as the increased free fatty acid levels in rapamycin-induced livers.

Increased free fatty acids may usually be caused by an enhanced lipolysis pathway, however, in rapamycin-induced fatty livers, the lipolysis enzyme ATGL was downregulated, and its inhibitor G0S2 was remarkably upregulated (Fig 4A and 4C) suggesting that increased free fatty acids are not from the enhanced lipolysis pathway. Collectively, the current study implies that although the fatty liver is an unwanted side effect of rapamycin, rapamycin-induced fatty livers may be less harmful than other types of fatty liver such as the high fat diet-induced fatty liver. Our observations may provide clues that support the broader application of rapamycin in the future. However, our obtained results can only reflect the situations in rapamycin-treated mice, which can be considered an important limitation of this study. Rapamycin's effects on human livers should be investigated in the future, such as liver functions and potential liver damage. The mechanisms of how rapamycin increases the interaction between p65 and IκBα and how rapamycin suppresses NLRP3 inflammasome activation are also important issues that need to be investigated in the future.

## Conclusion

Rapamycin suppresses NFκB nuclear translocation by enhancing the interaction between p65 and IκBα in rapamycin-induced fatty livers.

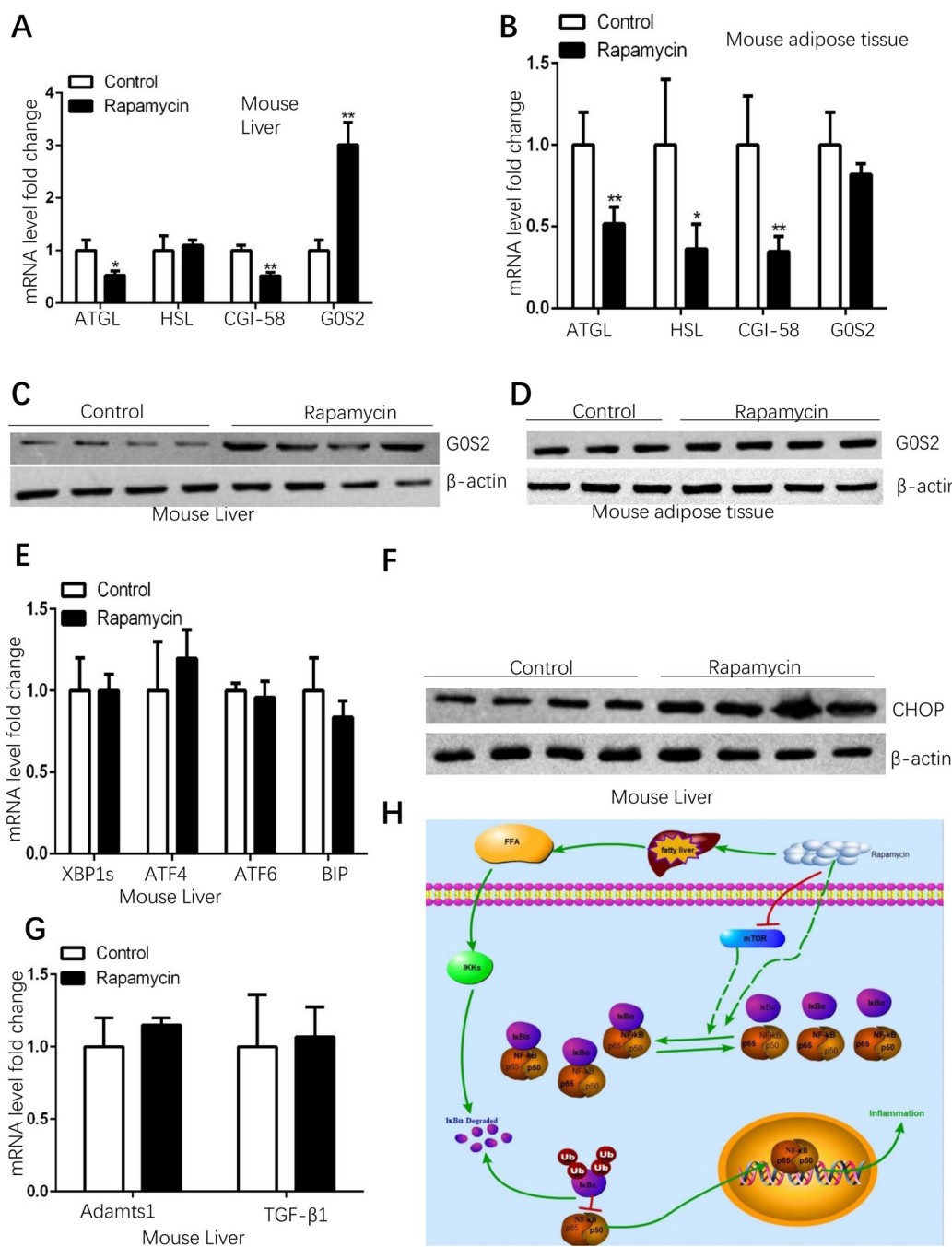

**Fig 4. Rapamycin inhibits the lipolysis pathway in rapamycin-induced fatty liver.** (Data represent mean ± SEM, **p < 0.01, *p<0.05). A. In rapamycin-induced fatty liver, the mRNAs of ATGL were downregulated and G0S2 was upregulated (Control group n = 7, rapamycin-treated group n = 8). B. In the adipose tissues of rapamycin-treated mice, G0S2 mRNA expression was not changed (Control group n = 7, rapamycin-treated group n = 8). C. G0S2 protein level was significantly increased in rapamycin-induced fatty livers (Control group n = 4, rapamycin group n = 4). D. Rapamycin treatment did not change G0S2 protein level in adipose tissue (Control group n = 3, rapamycin group n = 4). E. The mRNAs of ER stress-associated genes were not changed in rapamycin-induced fatty liver (Control group n = 7, rapamycin-treated group n = 8). F. CHOP was upregulated in rapamycin-induced fatty liver (Control group n = 4, rapamycin group n = 4). G. Eighteen weeks of rapamycin treatment did not increase liver fibrosis markers expression (Control group n = 7, rapamycin-treated group n = 8). H. Summary of the mechanisms that rapamycin suppresses inflammation level in rapamycin-induced fatty livers.

## Supporting information

**S1 Fig. Quantification of western blotting results.** A. Rapamycin decreases the phosphorylation level of S6 protein in mouse livers. B. Quantification of Fig 3A. C. Quantification of Fig 3B. D. Quantification of Fig 3C. E. Quantification of Fig 3D. F. Quantification of Fig 3F.
(TIF)

**S2 Fig. Quantification of western blotting results.** A. Quantification of Fig 4C. B. Quantification of Fig 4D. C. Quantification of Fig 4F. D. mRNA levels of key proteins which may affect lipid transport in livers. E. Rapamycin down-regulates DGAT2 mRNA expression level in mouse livers. F. Quantification of Fig 1B.
(TIF)

**S3 Fig. Western blotting results of IL-1β and Caspase-1 in rapamycin-induced fatty livers.** A. IL-1β protein levels were significantly decreased in rapamycin-induced fatty livers (IL-1β main band). B. Quantification of S3A Fig. C. Western blotting showed rapamycin dramatically reduced protein levels of cleaved IL-1β for secretion. D. Caspase-1 protein levels were significantly decreased in rapamycin-induced fatty livers (IL-1β main band). E. Quantification of S3D Fig. Rapamycin effects on protein levels of cleaved Caspase-1.
(TIF)

**S1 Table. Q-PCR primer sequences.**
(DOCX)

**S1 File. Original uncropped western blotting images.**
(PDF)

## Acknowledgments

We acknowledge all help, including both some experimental conditions and suggestions, from Dr. Weiwei Dang (Baylor College of Medicine, Houston, USA) and Dr. Zaiqing Yang (Huazhong Agriculture University, Wuhan, China) to facilitate our current investigation.

## Author Contributions

**Conceptualization:** An Yu.

**Data curation:** Chenliang Ge, Jiesheng Cui, Xingbo Dong.

**Formal analysis:** Changguo Ma, Luyang Sun.

**Funding acquisition:** An Yu.

**Investigation:** Chenliang Ge, Changguo Ma, Jiesheng Cui, Xingbo Dong, An Yu.

**Methodology:** Chenliang Ge, Changguo Ma, Jiesheng Cui, Xingbo Dong.

**Project administration:** Yanjiao Li.

**Resources:** Yanjiao Li.

**Software:** Luyang Sun.

**Supervision:** An Yu.

**Visualization:** Chenliang Ge, Changguo Ma.

**Writing – original draft:** An Yu.

**Writing – review & editing:** An Yu.

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
