## [Decision Letter · Decision Letter 0]

2 Dec 2022

PONE-D-22-30305Rapamycin suppresses inflammation and increases the interaction between p65 and IκBα in rapamycin-induced fatty liverPLOS ONE

Dear Dr. Yu,

Thank you for submitting your manuscript to PLOS ONE. After careful consideration, we feel that it has merit but does not fully meet PLOS ONE’s publication criteria as it currently stands. Therefore, we invite you to submit a revised version of the manuscript that provides the clarifications requested by the reviewer.

We look forward to receiving your revised manuscript.

Kind regards,

David M. Ojcius

Academic Editor

PLOS ONE

3. To comply with PLOS ONE submissions requirements, in your Methods section, please provide additional information regarding the experiments involving animals and ensure you have included details on (1) methods of sacrifice, (2) methods of anesthesia and/or analgesia, and (3) efforts to alleviate suffering.

“This work was supported by the funding of Yunnan Key Laboratory for Basic Research on Bone and Joint Diseases (BRBJD-2021-1) and the funding of Kunming University (XJ20220008).”

“This work was supported by the funding of Yunnan Key Laboratory for Basic Research on Bone and Joint Diseases (BRBJD-2021-1) and the funding of Kunming University (XJ20220008) awarded to An Yu. Yunnan Key Laboratory for Basic Research on Bone and Joint Diseases is a lab of Kunming University. The URL of Kunming University is https://www.kmu.edu.cn/  No, the sponsors did not play any role in the study design”

Reviewers' comments:

Reviewer's Responses to Questions

**Comments to the Author**

1. Is the manuscript technically sound, and do the data support the conclusions?

Reviewer #1: Yes

2. Has the statistical analysis been performed appropriately and rigorously? 

Reviewer #1: Yes

3. Have the authors made all data underlying the findings in their manuscript fully available?

Reviewer #1: Yes

4. Is the manuscript presented in an intelligible fashion and written in standard English?

Reviewer #1: No

5. Review Comments to the Author

Reviewer #1: The authors describe the effects of rapamycin in mice treated with rapamycin to induce fatty liver. The study appears to have been well performed and the conclusions are supported by the data. However, some questions came out during the review which requires clarifications and revisions.

Detailed comments:

1) The design of the study is quite counter-intuitive. How can a compound like rapamycin help reduce a symptom or disease that it caused in the first place? Why would people take rapamycin as an anti-aging drug if it induces fatty liver disease, which can affect quality of life and increase toxicity to a wide range of other substances due to poor liver function? Can the dose of rapamycin be reduced so that it does not induce fatty liver disease? How can we confirm that the first disease-causing exposure to rapamycin did not affect the subsequent treatment phase in terms of metabolism/phamacokinetics of the drug and induced drug resistance? Is the study design a relevant scenario for humans? Inflammation also contributes to fatty liver disease so this can hardly described as fatty liver disease as seen also by the absence of fibrosis.

2) What is the relevance for human use of injecting rapamycin intra-peritoneally? Does rapamycin induce liver disease orally?

3) Rapamycin does mimic some aspects of fasting, mainly due to its capacity to inhibit mTOR which is activated by food, especially proteins. The text lacks this kind of nuances in the description of the results.

4) In section 3.2, the authors conclude that rapamycin reduced mRNA levels of the NLRP3 inflammasome, but the results are incomplete since a two-hit mechanism is needed to induce its activation and cleavage of caspase-1 and IL-1beta. Further experiments are needed to make a conclusion about the effects on the NLRP inflammasome. For instance, the levels of cleaved caspase-1 and IL-1beta protein should be examined.

5) Previous studies have shown that free fatty acids can induce inflammation. In Fig. 2F, the authors show that rapamycin increases fatty acids in the liver, without inducing inflammation. The authors should present possible reasons to explain this paradox in the text.

6) The discussion should be re-written to present more insights from the results, comparisons with previous studies, possible limitations of the study, implications of the results, and/or future questions to be addressed.

7) Scale bars are missing on the microscopy images.

8) Some sentences should be rewritten for clarity. Grammar and typos should be checked throughout.

6. PLOS authors have the option to publish the peer review history of their article (what does this mean?). If published, this will include your full peer review and any attached files.

Reviewer #1: No

---

## [Author Response · Author response to Decision Letter 0]

23 Jan 2023

Response: We have modified our manuscript style according to your instructions!

Response: we would like to include all the original uncropped western blotting images in the Supporting Information. And we will indicate this point in our new cover letter.

3. To comply with PLOS ONE submissions requirements, in your Methods section, please provide additional information regarding the experiments involving animals and ensure you have included details on (1) methods of sacrifice, (2) methods of anesthesia and/or analgesia, and (3) efforts to alleviate suffering.

Response: we will include these pieces of information in our revised manuscript.

“This work was supported by the funding of Yunnan Key Laboratory for Basic Research on Bone and Joint Diseases (BRBJD-2021-1) and the funding of Kunming University (XJ20220008).”

“This work was supported by the funding of Yunnan Key Laboratory for Basic Research on Bone and Joint Diseases (BRBJD-2021-1) and the funding of Kunming University (XJ20220008) awarded to An Yu. Yunnan Key Laboratory for Basic Research on Bone and Joint Diseases is a lab of Kunming University. The URL of Kunming University is https://www.kmu.edu.cn/ No, the sponsors did not play any role in the study design”

Response: we will remove funding information from the Acknowledgments section. Thank you for your instructions.

Reviewers' comments:

Reviewer's Responses to Questions

Comments to the Author

1. Is the manuscript technically sound, and do the data support the conclusions?

Reviewer #1: Yes

2. Has the statistical analysis been performed appropriately and rigorously?

Reviewer #1: Yes

3. Have the authors made all data underlying the findings in their manuscript fully available?

Reviewer #1: Yes

4. Is the manuscript presented in an intelligible fashion and written in standard English?

Reviewer #1: No

5. Review Comments to the Author

Reviewer #1: The authors describe the effects of rapamycin in mice treated with rapamycin to induce fatty liver. The study appears to have been well performed and the conclusions are supported by the data. However, some questions came out during the review which requires clarifications and revisions.

Response: Dear reviewer, thank you very much for your careful work! We will try our best to address your concerns. 

Detailed comments:

1) The design of the study is quite counter-intuitive. How can a compound like rapamycin help reduce a symptom or disease that it caused in the first place? Why would people take rapamycin as an anti-aging drug if it induces fatty liver disease, which can affect quality of life and increase toxicity to a wide range of other substances due to poor liver function? Can the dose of rapamycin be reduced so that it does not induce fatty liver disease? How can we confirm that the first disease-causing exposure to rapamycin did not affect the subsequent treatment phase in terms of metabolism/phamacokinetics of the drug and induced drug resistance? Is the study design a relevant scenario for humans? Inflammation also contributes to fatty liver disease so this can hardly described as fatty liver disease as seen also by the absence of fibrosis.

Response: Yes, this study is kind of counter-intuitive, however, the design of this study is based on what has been observed regarding the effects of rapamycin on mice and even on humans. On one hand, rapamycin consistently extends the lifespans of mice in several studies (PMID: 19587680, 33145977, 35796299, 36179270, 23863708, 27549339); on the other hand, rapamycin treatment did cause fatty liver in mice (PMID: 23562079), and this phenotype is consistent with what we have observed in our experiments (Fig. 1A&B). Rapamycin has been proposed to prevent liver fibrosis or liver cirrhosis (PMID: 19790156, 10535884, 24966615, 27315465, 35405287); usually, liver fibrosis or liver cirrhosis is a devastating consequence of fatty liver, especially, alcohol-induced fatty liver and high fat diet-induced non-alcohol fatty liver, but rapamycin indeed induces fatty liver at the first place. One reasonable explanation for this paradox is that rapamycin-induced fatty liver is quite different from alcohol-induced fatty liver and high fat diet-induced fatty liver. The current study aims to distinguish rapamycin-induced fatty liver from other types of fatty livers such as a high fat diet-induced fatty lever, and we found that although the increased lipid accumulation and increased free fatty acid levels were observed in rapamycin-induced fatty liver, the expression levels of pro-inflammatory genes are even lower than that in the control group. This result indicated that rapamycin-induced fatty liver indeed is different from high fat diet-induced fatty liver, at least in the aspects of inflammation and fibrosis (Fig. 4G). Rapamycin is an FDA-approved drug that can be used to suppress Immunol-rejection after organ transplantations, yet a few people take rapamycin as an anti-aging drug due to the fact that rapamycin shows many slowing aging effects in a lot of pre-clinic studies, even though these studies also showed many side effects including fatty liver. These side effects indeed restrict broader applications of rapamycin, however, as long as people can prove that the side effects caused by rapamycin are less harmful than that was previously thought, or other drugs can cancel/reduce the side effects caused by rapamycin, then rapamycin might be considered a safer anti-aging drug that can be broadly used in the future. In addition, rapamycin does not necessarily cause side effects in all persons who use it, just like penicillin which saved many people’s lives from bacterial infections though it could be dangerous to people who are allergic to it. The key determinant of whether rapamycin can be broadly used as an anti-aging drug is if the anti-aging beneficial effects are much more than the detrimental side effects. One goal of the current study is to explain that fatty liver caused by rapamycin is less harmful than that was previously thought because rapamycin-induced fatty liver did not show increased inflammation levels (Figure 3 A&E) and did not increase the expression levels of fibrosis factors (Figure 4 G). The dose of rapamycin we used in this study is very similar to that used before, which has been proven competent to extend mouse lifespans (PMID: 20974732). If we reduced the rapamycin dose, which might be helpful to reduce fatty liver, we are not sure if the life-extending effect can be preserved due to the reduced dose. We cannot confirm that the first disease-causing exposure to rapamycin did not affect the subsequent treatment phase, and we also cannot exclude the effects of drug resistance, but as long as the final outcome of rapamycin treatment shows more beneficial effects than the adverse effects, then this drug is worthy to have more investigations in the future. Considering many studies in mice showed the life-extending effects of rapamycin, we assume that the final outcome of rapamycin treatment is a good one. The results of this study can be used as a reference for humans, but the results only reflect the situation of rapamycin treatment in mice; if we can gain funding with more money, we would happily like to organize a clinical trial referred to rapamycin side effects in the future. Yes, we can hardly term rapamycin-induced fatty liver as a disease, because it is more likely not to develop into liver fibrosis (Fig.4G) as other types of the fatty liver did, instead, we would more like to call it a symptom.

2) What is the relevance for human use of injecting rapamycin intra-peritoneally? Does rapamycin induce liver disease orally?

Response: The reason why we used intra-peritoneally injection of rapamycin in mice is that we can give each mouse rapamycin in an accurate dosage. If we mixed rapamycin in the chow diet, then the rapamycin amount that was received by each mouse will be variable due to the different food amounts that were taken by each mouse from day to day. Indeed, orally taking rapamycin caused dyslipidemia (PMID: 24654608) in some patients, and considering the liver is a major contributor to triglyceride levels in the blood, we assume that this result may imply that orally taking rapamycin in humans did affect lipid metabolism in the liver.

3) Rapamycin does mimic some aspects of fasting, mainly due to its capacity to inhibit mTOR which is activated by food, especially proteins. The text lacks this kind of nuances in the description of the results.

Response: Indeed, both fasting and rapamycin treatment are able to inhibit mTOR activity (PMID: 31406105), therefore rapamycin does mimic some aspects of fasting; however, in our current study, we found that rapamycin treatment and 24 h fasting induced very different gene expression patterns (Fig.1C). The reasons behind this phenomenon, we thought, are that rapamycin mainly affects mTOR signaling whereas fasting affects many pathways including Sirts (Sirt1-Sirt7), AMPK, and mTOR signaling. We will add these descriptions to the text.

4) In section 3.2, the authors conclude that rapamycin reduced mRNA levels of the NLRP3 inflammasome, but the results are incomplete since a two-hit mechanism is needed to induce its activation and cleavage of caspase-1 and IL-1beta. Further experiments are needed to make a conclusion about the effects on the NLRP inflammasome. For instance, the levels of cleaved caspase-1 and IL-1beta protein should be examined.

Response: We purchased Caspase-1 (cat# ab179515) and IL-1β (cat# ab234437) antibodies from Abcam and new western blotting experiments were added. The results of these western blotting experiments were put in Supplementary Figure 3. As has been shown in Supplementary Figure 3, 8-day rapamycin treatment significantly decreased IL-1β protein levels in mouse livers (Supplementary Figure 3 A&B), and notably, the cleaved IL-1β for secretion was hardly detectable in rapamycin-induced fatty livers, though another cleaved IL-1β protein levels were not affected by rapamycin (Supplementary Figure 3 C). Also, rapamycin treatment significantly decreased Caspase-1 protein levels (Supplementary Figure 3 D, E&F), though it is hard to determine whether cleaved Caspase-1 levels were affected by rapamycin due to the very low signaling (Supplementary Figure 3 F). These results indicate that NLRP3 inflammasome activation may be suppressed by rapamycin at least via decreasing Caspase-1 and IL-1β expression levels and reducing cleaved IL-1β form for secretion. 

5) Previous studies have shown that free fatty acids can induce inflammation. In Fig. 2F, the authors show that rapamycin increases fatty acids in the liver, without inducing inflammation. The authors should present possible reasons to explain this paradox in the text.

Response: our experimental results indicated that although increased free fatty acid levels were observed (Fig. 2F) and the upper stream of the NFκB pathway was activated (Fig. 3D) in rapamycin-induced fatty livers, the translocation of p65 into cellular nuclear was not increased (Fig. 3E), preventing that NFκB functions as a pro-inflammatory transcriptional factor. This is one of the reasonable explanations for the paradox. We would like to emphasize this point in the DISCUSSION part. 

6) The discussion should be re-written to present more insights from the results, comparisons with previous studies, possible limitations of the study, implications of the results, and/or future questions to be addressed.

Response: we have rewritten the discussion according to the reviewer’s suggestions.

7) Scale bars are missing on the microscopy images.

Response: Scale bars have been added now.

8) Some sentences should be rewritten for clarity. Grammar and typos should be checked throughout.

Response: We have had our manuscript checked and edited by an English native speaker.

---

## [Decision Letter · Decision Letter 1]

2 Feb 2023

Rapamycin suppresses inflammation and increases the interaction between p65 and IκBα in rapamycin-induced fatty liver

PONE-D-22-30305R1

Dear Dr. Yu,

We’re pleased to inform you that your manuscript has been judged scientifically suitable for publication and will be formally accepted for publication once it meets all outstanding technical requirements.

Kind regards,

David M. Ojcius

Academic Editor

PLOS ONE

Additional Editor Comments (optional):

Reviewers' comments:

Reviewer's Responses to Questions

**Comments to the Author**

1. If the authors have adequately addressed your comments raised in a previous round of review and you feel that this manuscript is now acceptable for publication, you may indicate that here to bypass the “Comments to the Author” section, enter your conflict of interest statement in the “Confidential to Editor” section, and submit your "Accept" recommendation.

Reviewer #1: All comments have been addressed

2. Is the manuscript technically sound, and do the data support the conclusions?

Reviewer #1: Yes

3. Has the statistical analysis been performed appropriately and rigorously? 

Reviewer #1: Yes

4. Have the authors made all data underlying the findings in their manuscript fully available?

Reviewer #1: Yes

5. Is the manuscript presented in an intelligible fashion and written in standard English?

Reviewer #1: Yes

6. Review Comments to the Author

Reviewer #1: (No Response)

7. PLOS authors have the option to publish the peer review history of their article (what does this mean?). If published, this will include your full peer review and any attached files.

Reviewer #1: No

---

## [Editor Report · Acceptance letter]

23 Feb 2023

PONE-D-22-30305R1 

Rapamycin suppresses inflammation and increases the interaction between p65 and IκBα in rapamycin-induced fatty livers 

Dear Dr. Yu:

I'm pleased to inform you that your manuscript has been deemed suitable for publication in PLOS ONE. Congratulations! Your manuscript is now with our production department. 

Kind regards, 

on behalf of

Dr. David M. Ojcius 

Academic Editor

PLOS ONE